# The Relationship between Healthy Vascular Aging with the Mediterranean Diet and Other Lifestyles in the Spanish Population: The EVA Study

**DOI:** 10.3390/nu16152565

**Published:** 2024-08-05

**Authors:** Leticia Gómez-Sánchez, David González-Falcon, Rocío Llamas-Ramos, María Cortés Rodríguez, Emiliano Rodríguez-Sánchez, Luis García-Ortiz, Inés Llamas-Ramos, Marta Gómez-Sánchez, Manuel A. Gómez-Marcos

**Affiliations:** 1Emergency Service, University Hospital of La Paz P. of Castellana, 261, 28046 Madrid, Spain; letici.gomez@salud.madrid.org; 2Primary Care Research Unit of Salamanca (APISAL), Health Centre of San Juan, Av. Portugal 83, 37005 Salamanca, Spain; gonzalezfalcondavid@gmail.com (D.G.-F.); rociollamas@usal.es (R.L.-R.); emiliano@usal.es (E.R.-S.); lgarciao@usal.es (L.G.-O.); martagmzsnchz@gmail.com (M.G.-S.); 3Institute of Biomedical Research of Salamanca (IBSAL), 37007 Salamanca, Spain; mariacortes@usal.es; 4Faculty of Nursing and Physiotherapy, Universidad de Salamanca, 37007 Salamanca, Spain; 5Department of Statistics, University of Salamanca, 37008 Salamanca, Spain; 6Department of Hematology, University Hospital of Salamanca, 37008 Salamanca, Spain; 7Primary Healthcare Management, Castilla y León Regional Health Authority (SACyL), 37007 Salamanca, Spain; 8Research Network on Chronicity, Primary Care and Health Promotion (RICAPPS), 37005 Salamanca, Spain; 9Department of Medicine, University of Salamanca, 37007 Salamanca, Spain; 10Department of Biomedical and Diagnostic Sciences, University of Salamanca, 37007 Salamanca, Spain; 11Home Hospitalization Service, Marqués of Valdecilla University Hospital, 39008 Santander, Spain

**Keywords:** Mediterranean diet, tobacco, alcohol, physical activity, sedentary time, healthy vascular aging

## Abstract

The aim of this study was to analyze the relationship between healthy vascular aging (HVA) and the Mediterranean diet alongside other lifestyles in a Spanish population aged 35 to 75 years without previous cardiovascular diseases. Methods: In this cross-sectional descriptive study, 501 individuals aged 35 to 75 years were recruited from five health centers by random sampling stratified by age and sex (55.90 ± 14.24 years, 49.70% men). HVA was determined in two steps. Step 1: Subjects with vascular damage to the carotid arteries or peripheral arterial disease were classified as non-HVA. Step 2: The study population was classified by age and sex using the percentiles of the vascular aging index (VAI), with VAI ≤p25 considered HVA and >p25 considered non-HVA. The VAI was estimated using the following formula (VAI = (log (1.09) × 10 cIMT + log (1.14) cfPWV) × 39.1 + 4.76. Carotid–femoral pulse wave velocity (cfPWV) was measured with the SphygmoCor^®^ device, and carotid intima–media thickness using Sonosite Micromax^®^ ultrasound. Mediterranean diet (MD) adherence, alcohol and tobacco use were recorded through validated questionnaires. Physical activity was assessed with the ActiGraph-GT3X^®^ accelerometer. Results: The mean VAI value was 61.23 ± 12.86 (men—63.47 ± 13.75 and women—59.04 ± 11.54; *p* < 0.001). HVA was found in 18.9% (men—19.9% and women—17.8%). In the multiple regression analysis after adjusting for possible confounding factors, the mean VAI value showed a positive association with alcohol use (β = 0.020) and sedentary hours per week (β = 0.109) and a negative association with hours of activity per week (β = −0.102) and with the number of healthy lifestyles (β = −0.640). In the logistic regression analysis, after adjusting for possible confounding factors and compared to those classified as non-HVA, subjects classified as HVA were more likely to show MD adherence (OR = 0.571), do more than 26 h per week of physical activity (OR = 1.735), spend under 142 h per week being sedentary (OR = 1.696), and have more than two healthy lifestyles (OR = 1.877). Conclusion: The results of this study suggest that the more time spent doing physical activity and the less time spent in a sedentary state, the lower the vascular aging index and the greater the likelihood of being classified in the group of subjects with HVA.

## 1. Introduction

In Western societies, population aging represents one of the biggest issues for sustainable development. Currently, 9% of the European population is aged over 65 years, and this is expected to rise to 25% by 2050 [1]. Vascular aging (VA) is characterized by the appearance of endothelial dysfunction and increased hardening of elastic arteries, an important risk factor for the development of age-related cardiovascular diseases. VA is mediated by excessive production of reactive oxygen molecules (ROS) and increased inflammation, causing reduced vasodilatory bioavailability of nitric oxide and remodeling of the arterial wall [2]. The rate of VA development depends on exposure to different factors damaging the arterial wall and the time of exposure to these factors, among which different lifestyles play an important role [3,4]. Thus, healthy lifestyle habits, such as regular aerobic exercise and certain components in the diet, coupled with dietary caloric restriction, are considered ‘frontline’ strategies for preventing and/or treating endothelial dysfunction linked to age [5].

There is currently no consensus on the best method to define VA. Methods to assess VA based only on arterial stiffness [6,7] are likely to be incomplete. Some authors consider that using the 10th and 90th or 25th and 75th percentiles of carotid–femoral pulse wave velocity (cfPWV) by age group to define a threshold value is more appropriate than establishing a fixed cut-off point (cfPWV > 10 m/s), given that percentile-based reference values are more accurate in identifying increased cardiovascular risk since cfPWV is influenced by age and sex. Other authors defend the analysis of arterial wall structure to define VA by measuring the intima–media thickness (IMT) of the carotid arteries, alongside arterial stiffness, as offering a more precise assessment of VA. Thus, the vascular aging index (VAI) published by Nilsson et al. [8] combines the IMT of the carotid artery and the cfPWV, two of the most widely used measures individually to estimate VA by reflecting arterial stiffness and subclinical atherosclerosis. This parameter was shown to predict cardiovascular events with good accuracy, providing complementary information in predicting the risk of cardiovascular diseases [8]. Considering all of the above, it is believed that the best definition of VA should be one that takes into account the following variables: age, blood pressure and sex (variables that have the greatest influence on arterial stiffness), as well as the cfPWV percentiles of the reference population studied (given the differences between populations) [9].

It has been shown that regular physical exercise (especially aerobic exercise) reduces arterial stiffness in the general population, thereby improving VA [5,10,11]. This occurs through increased availability of endothelial nitric oxide and a decrease in chronic low-grade vascular inflammation [3]. Moreover, numerous publications have described VA as being negatively associated with physical activity, and sedentary lifestyles have similarly been shown to have a positive link with arterial stiffness in adults [3,12,13]. While the association of VA with exercise was not found in the Framingham study [14], it was reported in a recent review [2] that aerobic exercise can reduce stiffness in the large arteries of middle-aged men and women. However, similar improvements through aerobic exercise training were not observed in estrogen-deficient postmenopausal women, suggesting that the effect differs with age between sexes [2].

The Mediterranean diet (MD) is the best known and most widely-researched dietary pattern worldwide. It is characterized by the consumption of a wide variety of foods, such as extra virgin olive oil, legumes, cereals, nuts, fruits, vegetables, dairy products, fish and wine [15,16]. Many of these foods provide various phytonutrients, among which polyphenols and vitamins play an important role [15,16]. It is currently one of the healthiest dietary patterns, probably due to the combination of many elements with antioxidant and anti-inflammatory properties [16,17,18,19]. These effects may be mediated by its beneficial composition of macronutrients (proteins, fats and carbohydrates) and micronutrients (vitamins and minerals) [20]. Numerous studies, systematic reviews and meta-analyses have established the protective effects of the Mediterranean diet against several chronic diseases, such as diabetes [21], obesity [22], cardiovascular diseases [18,23], cancer [19], aging disorders [24,25], cognitive impairment [17] and general mortality [26]. Furthermore, a recent review by the Cochrane Library [23] showed the benefits of the Mediterranean diet in the primary prevention of cardiovascular diseases, reducing mortality from ischemic heart disease, and improving the lipid profile, blood pressure, glycemia and adiposity. However, the level of evidence was weak to moderate and benefits in secondary prevention were not found [23]. Thus, the MD constitutes a useful tool in preventing cardiovascular diseases, making it one of the dietary patterns with the best relationship with cardiovascular diseases and other health outcomes [19,27,28]. However, studies that have analyzed the effect of diet on VA and endothelial dysfunction have focused more on certain nutrients and micronutrients than on dietary patterns, and the role that the MD pattern plays in endothelial dysfunction as a precursor to increased arterial stiffness and VA is unclear [20].

Smoking favors endothelial dysfunction by decreasing nitric oxide and increasing oxidative stress and by the production of proinflammatory cytokines that increase inflammation, vascular stiffness and, therefore, VA. Thus, the MESA study demonstrated that continuous smoking over a period of 10 years was associated with an increase in the aortic arch pulse wave velocity compared to non-smokers [29]. This increase occurred in men and in women. Numerous other publications have also described the positive association between VA and smoking [3,12,13]. In summary, while smoking accelerates VA and is closely related to atherosclerosis, smoking cessation can reduce the risk of vascular diseases. With regard to alcohol, some studies have shown a beneficial effect on VA of low or moderate alcohol consumption, particularly in the form of red wine [30,31]. However, excessive alcohol use is associated with an increase in arterial stiffness and a consequent increase in vascular aging [32,33].

Currently, it is not only the number of years lived that is valued but also their quality and the WHO thus defines healthy aging as “the process of developing and maintaining functional ability that enables well-being in older age” [34]. This ability is understood to include a person’s capacity to satisfy their basic needs, learn, grow and make decisions, walk unaided, and build or maintain relationships, thus contributing to a society that needs these outcomes. Achieving healthy aging must be focused on promoting healthy lifestyle habits that reduce the development of pathologies associated with aging in order to increase the quality of life of the aging population [34].

Few studies have so far jointly analyzed the effect of different lifestyles on healthy vascular aging (HVA) in the same population sample, and fewer have assessed physical activity and sedentary time as objectively measured with an accelerometer for a week. The main aim of this study is, therefore, to analyze the relationship between HVA with different lifestyles in Caucasian subjects aged between 35 and 75 without previous cardiovascular disease, with a secondary aim of analyzing sex differences.

## 2. Materials and Methods

### 2.1. Study Design

This cross-sectional descriptive study analyzes data from the 501 subjects included in the EVA study Association between different risk factors and the accelerated vascular aging study [35] registered at ClinicalTrials.gov. Identifier NCT02623894.

### 2.2. Study Population

The subjects were selected following random sampling with replacement stratified by age group (35, 45, 55, 65 and 75 years) and by sex. A total of 501 subjects were selected from 5 urban health centers, with 100 subjects from each age group (50 men and 50 women). The reference population was the 43,946 subjects in the individual health card database. The inclusion of subjects in the study was carried out between June 2016 and November 2017. Inclusion criteria were being aged between 35 and 75 years and having signed the informed consent. Exclusion criteria were having a terminal illness or not being able to travel to the research unit for any other reason, having a history of cardiovascular disease or a glomerular filtration rate < 30 mL/min/1.73 m^2^, being diagnosed with a chronic inflammatory disease or any acute inflammatory process in the last three months or being treated with estrogen, testosterone or growth hormone.

Figure 1 is a flow diagram of the subjects included, excluded and the causes by age and sex groups, as well as the reference population.

### 2.3. Variables and Measuring Instruments

A detailed description of the study methodology, together with the inclusion and exclusion criteria, can be found in the study protocols [35].

#### 2.3.1. Healthy Lifestyles

##### Mediterranean Diet

Adherence to the MD was assessed using the PREDIMED study questionnaire [36]. Adherence to the MD was assessed using the PREDIMED study questionnaire with 7447 asymptomatic participants aged between 55 and 80 years from the Spanish population with a high risk of coronary heart disease [36]. It consists of 14 items, of which 12 ask about the frequency of food consumption and 2 about the typical eating habits of the Spanish population. It was validated again in 2021 with 6760 subjects in the PREDIMED-Plus study (n = 6760, 55–75 years) [37], but it has not been validated for the population aged under 55 years. Each question is scored zero or one, with one point scored for use of olive oil as the main cooking fat, daily consumption of four or more tablespoons of olive oil (one tablespoon = 13.5 g), two or more servings of vegetables, three or more pieces of fruit, less than one serving of red or processed meat, less than one serving of animal fat, less than a 100 mL cup of sugary drinks, eating white meat in greater proportion to red meat, weekly consumption of seven or more glasses of wine, three or more servings of legumes, three or more servings of fish, three or more servings of nuts or dried fruits, two or more servings of sofrito (home-made sauce of onions and/or garlic and tomato, slow-fried in extra-virgin olive oil) and less than two baked goods. The final score ranges between 0 and 14 points, with MD adherence considered for a score above the median (7 or more points) [36]. There is no consensus on this cut-off point, which thus represents a limitation.

##### Alcohol and Smoking

Alcohol consumption was recorded using a standardized questionnaire (recording type and amount of alcohol ingested during a week, in g/week). Adequate alcohol consumption for women was considered to be <140 g/week and for men <210 g/week [38].

Smoking was assessed with the questionnaire used in the Monica study [39], recording whether the participant was a smoker or not, the number of cigarettes per day and the years they had been smoking.

##### Physical Activity and Sedentary Time

Physical activity was objectively assessed using the validated ActiGraph-GT3X^®^ accelerometer (ActiGraph, Shalimar, FL, USA) [40]. The original data from the accelerometers were collected at a frequency of 30 Hz. The specific requirements for their use were as follows: (a) the accelerometers were fixed at the waist and placed on the axillary line at the level of the iliac crest of the hip right or left; (b) the accelerometer was used for seven consecutive days, except when bathing or swimming. Data were invalid if the number of days was <3 days per week or the wear time was <8 h per day. Data were recorded at 1 min intervals. The accelerometers recorded the time of physical activity performed, measured in hours per week and sedentary time per week.

#### 2.3.2. Assessment of Vascular Structure, Function and Vascular Aging

##### Assessment of Intima–Media Thickness

The intima–media thickness (IMT) of the carotid arteries was performed by two trained researchers, with intraclass and intraobserver correlation coefficient values of 0.974 and 0.897 for interobserver agreement in the measurements made in 20 subjects before starting the study. The device used was the Sonosite Micromax^®^ ultrasound machine (Sonosite Inc., Bothell, WA, USA), with a 5–10 MHz multifrequency high-resolution linear transducer and Sonocal software 1 845-747-0485 (Washington, DC, USA) that automatically measures the IMT of carotid arteries. The IMT measurements were carried out following the protocol published by our research group [41].

##### The Ankle–Brachial Index

The ankle–brachial index was measured using the VaSera VS-1500^®^ device (Fukuda Denshi, Tokyo, Japan). The presence of vascular injury was established following the criteria included in the clinical practice guidelines [42].

##### Assessment of Carotid–Femoral Pulse Wave Velocity

(cfPWV) was analyzed with the patient in the supine position using the SphygmoCor^®^ device (AtCor Medical Pty Ltd., Head Office, West Ryde, Australia), which calculates cfPWV by estimating the time delay with respect to the R wave of the electrocardiogram. To do this, the distance between the sternal notch and the point where the sensor is placed on the carotid and femoral arteries was determined by means of a measuring tape [43]. The quality of the pulse wave is established by the device-specific software, and the manufacturer’s instructions were followed at all times to estimate the cfPWV.

##### Vascular Aging Index (VAI)

The VAI was estimated with the following formula [28]:VAI = (log (1.09) × 10 cIMT + log (1.14) × aPWV) × 39.1 + 4.76
where cIMT is the carotid intima–media thickness, aPWV is the aortic pulse wave velocity equivalent to cfPWV, and log is the natural logarithm with base e. VAI is a parameter that combines methods to measure different arterial properties. It takes into account the IMT-assessed vascular structure of the carotid arteries, which reflects already established atherosclerosis, and the equivalent of aPWV to cfPWV, which reflects arterial stiffness [42].

#### 2.3.3. Definition of Healthy Vascular Aging

Healthy vascular aging (HVA) was determined in two steps. Step 1: Subjects with vascular injury in the carotid arteries or peripheral arterial disease were classified as non-HVA. Step 2: The studied population was classified by age and sex using the VAI percentiles [8]. Subjects with values ≤ 25th percentile were classified as HVA, and subjects with values > 25th percentile were classified as non-HVA.

#### 2.3.4. Anthropometric Measurements and Cardiovascular Risk Factors

Blood pressure, weight and height were measured at the Primary Care Research Unit of Salamanca (APISAL) following the recommendations published in the study protocol [35]. Subjects were considered to have hypertension if they were taking antihypertensive drugs or with blood pressure values ≥ 140/90 mmHg; to have type 2 diabetes mellitus if they were taking hypoglycemic agents or had fasting plasma glucose values ≥ 126 mg/dL or HbA1c ≥ 6.5%; and to have dyslipidemia if they were taking lipid-lowering agents or had fasting total cholesterol values ≥ 240 mg/dL, low-density lipoprotein cholesterol (LDL-C) ≥ 160 mg/dL, high-density lipoprotein cholesterol (HDL-C) ≤ 40 mg/dL in men and ≤50 mg/dL in women, or triglycerides ≥ 150 mg/dL. Obesity was defined as having a BMI ≥ 30 kg/m^2^ [41].

#### 2.3.5. Analytical Tests

Venous blood samples were taken between 08:00 and 09:00 h after subjects had fasted for 12 h. Fasting plasma glucose, serum total cholesterol, high-density lipoprotein cholesterol and triglyceride levels were measured using a standard enzyme. Glycosylated hemoglobin was measured with an automated immunoturbidimetric assay method. All analytical tests were processed in the same laboratory [35].

### 2.4. Statistical Analysis

Data are presented using means ± standard deviations and numbers or percentages depending on whether variables are continuous or categorical. The comparison between men and women was carried out with chi-square tests for percentages and Student’s *t* tests for continuous variables. The Pearson correlation coefficient was used to analyze the relationship between continuous variables.

The constant variance hypothesis was verified with variance homogeneity tests using the Levene statistic in all variables analyzed. Using the classification of HVA or non-HVA as the dependent variable, the different lifestyles as independent variables and the age and cardiovascular risk factors (mean arterial pressure, atherogenic index, HbA1c and body mass index) as adjustment variables, the *p* value was >0.050 in all the variables. The outlier values were analyzed with box–whisker plots; the number of values was not excluded from the analysis since, in those variables with outlier values, the number is small and within biological values.

To analyze the association between the average VAI score and the different lifestyles, six multiple linear regression models were applied, using the VAI as dependent variable and the number of healthy lifestyles, average MD adherence score, weekly alcohol consumption in g/week, years of smoking, number of active hours per week and number of sedentary hours per week as independent variables.

To analyze the association between HVA individuals and different lifestyles, six logistic regression models were used. Those classified as HVA or not HVA were dependent variables (coded HVA = 1, no HVA = 0), and healthy lifestyles (coded yes = 1, No = 0) were independent variables. The median score for the different healthy lifestyles was used, except in the case of alcohol, where having adequate alcohol consumption (considered to be <140 g/week in women and 210 g/week in men) or not was used. We considered a healthy lifestyle to be when subjects had (a) adherence to the Mediterranean diet score greater than 7; (b) adequate alcohol consumption for women was considered to be <140 g/week and for men < 210 g/week; (c) no smoking; (d) more than 26 h a week of physical activity; € less than 142 h a week of sedentary time; and (f) more than two healthy lifestyles.

In all models, age, sex, mean arterial pressure, atherogenic index, HbA1c and body mass index were included as adjustment variables. The selection criteria for these were one for each cardiovascular risk factor with a *p* value < 0.05 in Table 2, which combined the largest number of variables related to said factor.

All analyses were performed overall and by sex. The SPSS Statistics program for Windows, version 28.0 (IBM Corp, Armonk, NY, USA) was used. A value of *p* < 0.05 was considered the statistical significance limit.

### 2.5. Ethical Principles

This EVA Project was approved by the Drug Research Ethics Committee of the Salamanca health area on 4 May 2015 (CEIm reference code PI 15/01039). Before the start of the study, all participants signed the informed consent. During the course of the study, the standards of the Declaration of Helsinki [44] and the WHO guidelines for observational studies were followed. Subject confidentiality was guaranteed at all times, in accordance with the provisions of Organic Law 3/2018, of December 5, on personal data protection and the guarantee of digital rights, and European Parliament Regulation (EU) 2016/679 and the Data Protection Council of 27 April 2016 (GDPR).

## 3. Results

### 3.1. Lifestyles, Risk Factors, and Vascular Structure and Function of the Subjects Included, Overall and by Sex

In the overall analysis, the mean MD score was 7.15 ± 2.07, with adherence to the MD of 42.7%. Average alcohol consumption was 40.47 ± 63.15 g per week, and adequate consumption was 50.1%. The years of smoking average was 12.98 ± 17.41, and the percentage of subjects who had never smoked was 73.3%. The hours of physical activity per week was 27.09 ± 9.52, with 49.7% doing more than 26 h per week. Mean sedentary time was 140.75 ± 9.56 h per week, with 51.1% being sedentary for under 142 h per week. The average number of healthy lifestyles was 2.66 ± 1.29, and 50.5% had more than 2 healthy lifestyles. In the analysis by sex, women showed higher values of MD, total physical activity and the average number of healthy lifestyles, and less alcohol consumption and sedentary time per week than men. In the overall analysis, the mean value for IMT was 0.682 ± 0.109 mm, cfPWV was 6.53 ± 2.03 m/s, and the VAI was 61.23 ± 12.86. These values were higher in men than in women. Table 1 presents the values of the different cardiovascular risk factors as well as the drugs used in hypertension, dyslipidemia and type 2 diabetes mellitus treatment, overall and by sex.

The percentage of subjects according to the number of healthy lifestyles is shown overall in Figure 2 and by sex in Figure 3. The percentage of subjects who presented at least two healthy lifestyles was 50.3% (men: 40.3%; women: 59.7%; *p* < 0.001).

### 3.2. Lifestyles, Risk Factors, and Vascular Structure and Function According to Vascular Aging

Table 2 shows the mean values for lifestyles, cardiovascular risk factors and IMT, cfPWV and VAI parameters in subjects with HVA and without HVA. Total physical activity was 29.19 vs. 26.58 h per week; sedentary hours per week was 138 vs. 141, the percentage of subjects who had never smoked was 69.1% vs. 67.0% and the percentage of subjects with more than 2 healthy lifestyles was 60.6% vs. 48.3% in subjects classified as HVA compared to those classified as non-HVA (*p* < 0.05). All cardiovascular risk factors, except total cholesterol and LDL cholesterol, showed more favorable values in subjects classified as HVA. The parameters of vascular structure and function and the VAI showed similar results.

Figure 4 represents the differences in the VAI, cfPWV and cIMT values between subjects with healthy lifestyles and without healthy lifestyles, overall and by sex.

### 3.3. Correlation of Vascular Aging with Mediterranean Diet and Other Lifestyles

Table 3 shows the correlation of the VAI with lifestyles adjusted by age, overall and by sex. The VAI showed a negative association with the mean value of MD (r = −0.102), hours of total physical activity (r = −0.158) and the number of healthy lifestyles (r = −0.199), and a positive association with consumption of alcohol (r = 0.228), years of smoking (r = 0.092) and sedentary time (r = 0.165).

### 3.4. Association between Vascular Aging Index and Healthy Lifestyles: Multiple Regression Analysis

In multiple regression analysis after adjusting for potential confounders, the mean VAI value showed a negative association with hours of total physical activity per week (β = −0.102, 95% CI: −0.176 to −0.028) and with the number of healthy lifestyles (β = −0.640, 95% CI: −1.195 to −0.086) and a positive association with alcohol consumption (β = 0.020, 95% CI: 0.008 to 0.032) and with sedentary hours per week (β = 0.109; 95% CI: 0.036 to 0.183). The analysis by sex yielded similar results, as shown in Table 4, except for alcohol consumption and the number of healthy lifestyles, which were significant in men but not in women.

### 3.5. Association between Vascular Aging Index and Healthy Lifestyles: Logistic Regression Analysis

After adjusting for possible confounding factors, the logistic regression analysis yielded the following results: compared with non-adherence to the Mediterranean diet, adherence to the Mediterranean diet was associated with a higher likelihood of being classified as HVA, relative to non-HVA (OR = 0.571; 95% CI: 0.333 to 0.981); compared with less than 26 h per week of physical activity, doing more than 26 h per week of physical activity was associated with a higher likelihood of being classified as HVA, relative to non-HVA (OR = 1.735; 95% CI: 1.048 to 2.871); compared with more than 142 h per week of sedentary time, spending less than 142 h per week being sedentary was associated with a higher likelihood of being classified as HVA, relative to non-HVA (OR = 1.696; 95% CI: 1.025 to 2.805); and compared to having less than two healthy lifestyles, more than two healthy lifestyles was associated with a higher likelihood of being classified as HVA, compared to non-HVA (OR = 1.877; 95% CI: 1.123 to 3.136). In the analysis by sex, the results were similar, although there were no significant differences, except for smoking in men, as reflected in Table 5.

## 4. Discussion

To our knowledge, this is the first study to analyze the relationship between estimated HVA and VAI with different lifestyles in a Spanish adult population without a history of cardiovascular disease. The main findings of this study are as follows: (1) using the VAI as an aging criterion, approximately one in five subjects is classified as HVA, and (2) in both multiple and logistic regression analyses, the lifestyles most strongly associated with the VAI and with HVA is the amount of physical activity and sedentary time, measured objectively in hours/week with an accelerometer over seven days.

In line with results published by other authors, this study shows that the more hours of physical activity and the fewer hours of sedentary activity, the lower the VA. Thus, numerous studies have shown a negative association of arterial stiffness with physical activity and a positive association with a sedentary lifestyle in adults [3,12,13], with aerobic exercise, in particular, improving VA [5,10,11], and longitudinal studies have shown that the increase in arterial stiffness over five years, measured with the cfPWV, is linked to time spent doing physical activity and sedentary time [45]. Ahmadi et al. [46], for example, found that limiting sedentary time was associated with slower progression of aortic stiffness. However, this association has not been found in all studies, as in the case of the Framingham study [14], probably because physical activity was not objectively measured with an accelerometer in that study. On the other hand, a recent review [2] shows that aerobic exercise improves the stiffness of large arteries in middle-aged men and women, although the same effects were not observed in postmenopausal women with estrogen deficiency, suggesting that the effect may differ with age between sexes [2]. Supporting the results of this present study, other authors have found that time spent watching television is related to worse cardiovascular health [47,48], and Wennman et al. [49] found that the risk of cardiovascular disease was higher in men who spent four hours or more watching television and in women watching two to three hours. Therefore, taking into account the benefits of physical activity [50] and the dangers of a sedentary lifestyle [51], it seems logical that HVA improves with increased physical activity and decreased sedentary time. Nevertheless, sedentary lifestyles are becoming more prevalent worldwide due to the lack of available spaces for exercise, the increase in sedentary occupational behaviors such as office work, and the greater usage of electronic devices, television or video [52]. Indeed, studies carried out in Europe, the United States and Australia have shown that adults spend half of their working day (4.2 h/day) and around 2.9 h/day of their free time sitting [53].

As in previous studies, MD adherence was higher in women [54]. Contrary to what may be expected, the mean MD score did not show an association with the VAI, either overall or by sex. However, the overall analysis showed an OR of 0.571 in logistic regression after adjusting for possible confounding factors. Similarly, the increase in cfPWV over 5 years has not been associated with the MD [45]. Nevertheless, there are several studies that show subjects with adherence to an MD pattern having improved endothelial function, inflammation, oxidative stress or other conditions that predispose people to cardiovascular events, such as obesity or type 2 diabetes mellitus, suggesting that it may play an important role in HVA by preventing the onset of certain diseases and improving the VA process [20,55]. Thus, some authors consider that dietary patterns such as MD should be considered ‘frontline’ strategies in preventing and/or treating age-related cardiovascular dysfunction [5]. However, it should not be forgotten that, as shown in a recent review [25], the majority of studies carried out have focused on the benefits of certain components of MD in specific processes of general aging without specifically analyzing the effect on VA or HVA, concluding that the benefits of the MD become more important in older people, especially for achieving healthy aging. Different studies have thus shown the benefits of magnesium and potassium in preventing the reduction in muscle mass [56] and improved bone and mineral homeostasis with diets rich in calcium, vitamins D, A, B, C, E and minerals such as potassium and magnesium, thereby modulating long-term bone health [57], improved bone density and prevention of osteoporosis in Spanish postmenopausal women [58], and increased immune responses through vitamin D, E and probiotic intake [59]. Given all of the above, although there is sufficient evidence of the MD pattern having general health benefits, questions remain about whether it is the MD pattern or certain micronutrients and vitamins that are responsible for HVA and whether the effect is the same in all the age groups and in both sexes.

In summary, this paradoxical finding in our study may have several explanations: 1. The influence of the Mediterranean dietary pattern on HVA may not be so strong, with micronutrients and vitamins playing an important role, as shown by different studies carried out in animals or in humans [60,61,62,63,64]. 2. Most of the studies assessing the benefits of the Mediterranean diet have been carried out on specific diseases or factors and not on their importance in vascular aging. 3. The age range of our study’s population is broad and we should not forget that the MEDAS questionnaire has not been validated in people under 55 years of age, and there is no consensus on the cut-off points for good adherence. 4. Adherence to the Mediterranean diet is greater in older subjects, and the importance of age in aging is probably greater than that of the Mediterranean diet. Finally, there may be confounding factors that we have not considered in this work.

Therefore, more research is needed to provide a better understanding of the mechanism of how the MD acts on aging, although on the basis of the current evidence, the MD could be recommended as an anti-aging therapy to prevent frailty and maintain functionality until the final stages of life [24].

In this study, alcohol consumption measured in grams per week was positively associated with the VAI overall and in men. These results are in line with those shown by Hwang et al. [65], and data published in a review in 2022 [32] analyzing the relationship between arterial stiffness and alcohol consumption concluded that greater alcohol consumption is associated with worse arterial stiffness values, using different measurement methods (oscillometry or tonometry) and obtaining similar results across the different populations studied (European, American and Asian) and for almost all types of alcoholic beverages consumed. Some studies only show an association between alcohol intake and cfPWV in men [32,33]. This association was not found in our logistic regression analysis and is a result that is not consistent with what has been published by several authors who have suggested a J-shaped relationship between alcohol consumption and arterial stiffness measured with cfPWV [33,66]. The discrepancies between this study and others may be explained by different reasons. The VAI used as a criterion is an aging index that, in addition to arterial stiffness, takes into account the thickness (IMT) of the carotid artery. Furthermore, we cannot forget that the studies analyzed were heterogeneous, using different methods to measure arterial stiffness, with alcohol consumption recorded subjectively, and the majority were cross-sectional studies. All of this, together with the fact that many of the studies have methodological weaknesses and may overestimate the benefits of alcohol on cardiovascular diseases, makes the results inconclusive [67,68,69]. Therefore, the debate on the relationship between alcohol consumption and VA remains an open one, given that the relationship is complex and potentially affected by several factors such as the type of alcohol, consumption levels, sex and age differences.

This study did not find an association between years of smoking and VAI, in line with some other studies that did not yield significant changes in the chronic effect of smoking on arterial stiffness [32]. However, most studies, as reflected in the results of the Gutenberg Health Study published in 2023, have concluded that chronic smoking is strongly and dose-dependently associated with an increase in arterial stiffness, regardless of sex, but with an association that is stronger in men [70], although they used the stiffness index and the increase index as measures of stiffness. On the other hand, in the logistic regression analysis, we only found an association in men, in line with the results published by other authors [71], suggesting different patterns of vascular system stiffness in men and women due to smoking.

The results of this study suggest that the greater the number of healthy lifestyles, the greater the probability of being classified in the group of subjects with HVA and the lower the VAI value. This is likely due to the fact that such lifestyles mutually enhance the beneficial effect of HVA. Thus, the WHO considers that in order to achieve healthy aging, healthy lifestyle habits that reduce the development of pathologies associated with aging must be promoted in order to increase the quality of life of the aging population. It also recognizes that evidence is scarce, and that research on healthy aging should be promoted, with health systems adapting to the needs of older people, generating the human resources necessary to be able to carry out comprehensive care for older people [34]. In the review by Claas et al. [72], lifestyles such as adequate dietary intake and regular physical activity are seen as essential in primary prevention to avoid cardiovascular risk factors that will determine vascular aging. These measures, together with the absence of smoking and certain behavioral factors such as stress control and sleep duration, should be considered in lifestyle modification programs, and these healthy lifestyles should be implanted from childhood [73].

Nevertheless, we cannot forget that vascular aging is a complex phenomenon in which many factors intervene, some of which are beyond our control, such as age, sex and the genetic load of each individual. Modifiable risk factors are those that we acquire throughout life, conditioned by different lifestyles that influence individual cardiovascular risk factors and psychological and inflammatory factors, which cause an increase in oxidative stress and inflammation, leading to endothelial dysfunction. Thus, it is known that arterial aging is associated with changes that impact vascular function [3,4]. However, the extent to which these changes are produced by the action of environmental factors, lifestyle, psychological factors, inflammation or oxidative stress, predisposing each individual to mark a rhythm in his own vascular aging, has not been sufficiently researched. In this study we have tried to control for most of the confounding factors. Firstly, the sample was selected randomly, but it only represents the urban population. As adjustment variables for the cardiovascular risk factors, we tried to use those covariates that may best represent each of the cardiovascular risk factors, although we did not include inflammatory and psychological factors as possible confounding factors. Furthermore, three of the four lifestyles analyzed were recorded through questionnaires, so there may be some information bias. Finally, we must emphasize that physical activity and sedentary lifestyle, the only lifestyle objectively assessed with accelerometry, is the lifestyle that shows a clear and consistent association with vascular aging. For all these reasons, the results must be interpreted with caution, taking into account that there may be spurious associations that we have not considered.

Finally, hormonal and non-hormonal factors influence differences between the sexes. The protection of endogenous estrogen until menopause in women is well known. Furthermore, in males, arterial stiffness increases linearly from puberty, which indicates that females intrinsically have stiffer main arteries than males, effects that are mitigated by sex steroids during reproductive life. Other factors, such as height, body fat distribution and inflammatory factors, may also play a role [74,75].

### Limitations and Strengths

This study has a number of limitations and strengths. The main limitation is that the analysis is based on cross-sectional data, which prevents us from establishing causality. Another limitation is that three of the four lifestyles analyzed were recorded subjectively through questionnaires, and the MD adherence questionnaire has not been validated in the Spanish population under 55 years of age. Among the strengths of the study is sample selection through random sampling stratified by age and sex groups from a reference population of more than 43,000 subjects, while the measurement of IMT of carotid arteries and cfPWV was performed under standardized conditions by experienced evaluators and with validated devices. Furthermore, all analytical measurements were carried out in laboratories with adequate quality controls.

## 5. Conclusions

The results of this study suggest that the longer we do physical activity and the less time we spend sitting, the lower the rate of vascular aging and the greater the likelihood of being classified in the group of subjects showing HVA.

## Figures and Tables

**Figure 1 nutrients-16-02565-f001:**
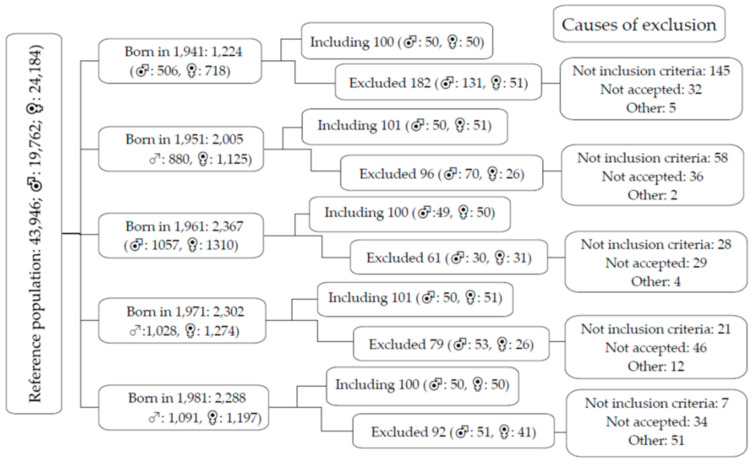
Flow diagram of the study, indicating reference population by age decade, overall, and by gender, subjects included and excluded, and the main causes of exclusion.

**Figure 2 nutrients-16-02565-f002:**
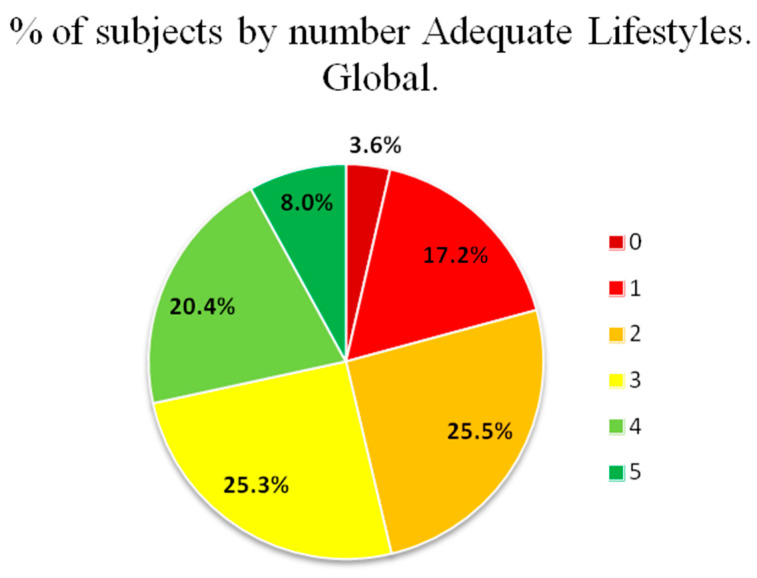
Percentage of subjects overall according to the number of healthy lifestyles.

**Figure 3 nutrients-16-02565-f003:**
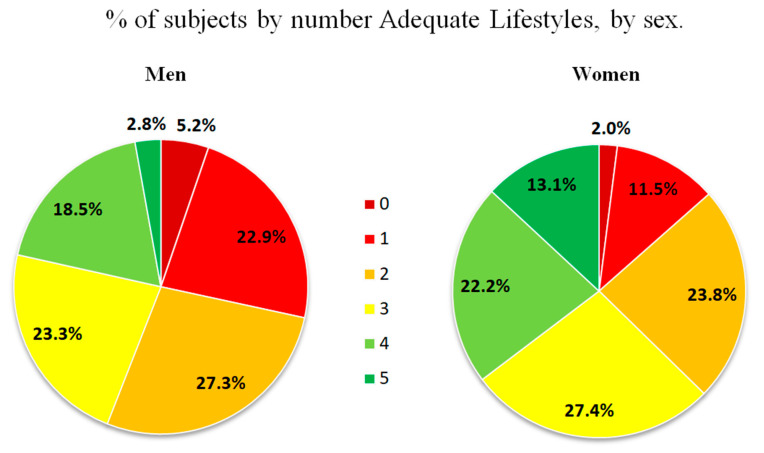
Percentage of subjects by sex according to the number of healthy lifestyles.

**Figure 4 nutrients-16-02565-f004:**
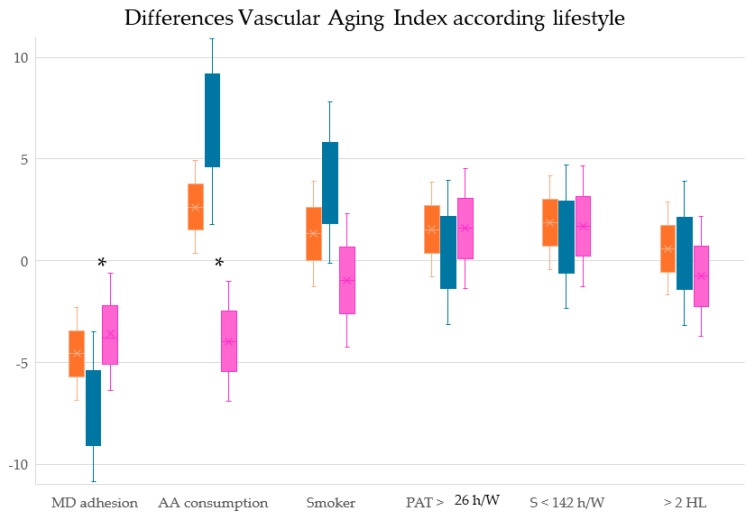
Differences in VAI scores between subjects with and without healthy lifestyles, overall and by sex. Healthy lifestyles were considered to be the following: (a) adherence to the Mediterranean diet score greater than 7; (b) adequate alcohol consumption for women was considered to be <140 g/week and for men <210 g/week; (c) no smoking; (d) more than 26 h a week of physical activity; (e) less than 142 h a week sitting; and (f) more than two healthy lifestyles. MD, Mediterranean diet; AA, adequate alcohol consumption; PAT h/W, physical activity total hours/week; S h/W, sedentary hours per week; HL, number of healthy lifestyles; VAI, vascular aging index. Brown represents the overall results, blue the results for men and pink the results for women. * reflects whether the difference between subjects with and without healthy lifestyles is significant.

**Table 1 nutrients-16-02565-t001:** Lifestyles, risk factors, and vascular structure and function of the subjects included overall and by sex.

Lifestyles	Overall (501)	Men (249)	Women (252)	*p* Value
MD score	7.15 ± 2.07	6.68 ± 1.97	7.60 ± 2.08	<0.001
Adherence to MD, n (%)	214, 42.7%	89, 35.7%	125, 49.6%	0.001
Alcohol consumption (g/W)	40.47 ± 63.15	61.54 ± 74.65	19.64 ± 39.54	<0.001
Adequate alcohol consumption, n (%)	251, 50.1%	94, 37.8%	157, 62.3%	<0.001
Years of smoking	12.98 ± 17.41	14.43 ± 18.91	11.55 ± 15.69	0.064
Never smoked, n (%)	367, 73.3%	183, 73.5%	184, 73.0%	0.920
Total physical activity (h/W)	27.09 ± 9.52	26.01 ± 9.51	28.17 ± 9.43	0.011
More than 26 h/W, n (%)	246, 49.7%	107, 43.1%	139, 56.3%	0.004
Sedentary time (h/W)	140.75 ± 9.56	141.78 ± 9.57	139.72 ± 9.45	0.017
Less than 142 h/W, n (%)	253, 51.1%	113, 45.6%	140, 56.7%	0.015
Number of healthy lifestyles	2.66 ± 1.29	2.35 ± 1.24	2.96 ± 1.28	<0.001
More than 2 healthy lifestyles, n (%)	253, 50.5%	102, 41.0%	151, 59.9%	<0.001
*Conventional risk factors*				
Age (years)	55.90 ± 14.24	55.95 ± 14.31	55.85 ± 14.19	0.935
Systolic blood pressure (mmHg)	120.69 ± 23.13	126.47 ± 19.52	114.99 ± 24.96	<0.001
Diastolic blood pressure (mmHg)	75.53 ± 10.10	77.40 ± 9.38	73.67, ± 10.46	<0.001
Mean arterial pressure (mmHg)	90.58 ± 12.61	93.76 ± 11.13	87.44 ± 13.21	<0.001
Pulse pressure (mmHg)	45.17 ± 19.81	49.06 ± 16.68	41.31 ± 21.83	<0.001
Hypertensive, n (%)	147, 29.3%	82, 32.9%	65, 25.8%	0.095
Antihypertensive, n (%)	96, 19.2%	50, 20.1%	46, 18.3%	0.650
Total cholesterol (mg/dL)	194.76 ± 32.49	192.61 ± 32.26	196.88 ± 32.65	0.142
LDL cholesterol (mg/dL)	115.51 ± 29.37	117.43 ± 30.12	113.61 ± 28.54	0.148
HDL cholesterol (mg/dL)	58.75 ± 16.16	53.19 ± 14.12	64.22 ± 16.20	<0.001
Triglycerides (mg/dL)	103.06 ± 53.20	112.28 ± 54.40	93.95 ± 50.46	<0.001
Atherogenic index	3.53 ± 1.07	3.84 ± 1.12	3.24 ± 0.93	<0.001
Dyslipidemic	191, 38.1%	95 ± 38.2%	96, 38.1%	98.9%
Lipid-lowering, n (%)	102, 20.4%	49, 19.7%	53, 21.0%	0.740
Plasma glucose (mg/dL)	88.21 ± 17.37	90.14 ± 18.71	86.30 ± 15.73	0.013
HbA1c (%)	5.49 ± 0.56	5.54 ± 0.63	5.44 ± 0.47	0.043
Diabetes mellitus type 2, n (%)	38, 7.6%	26, 10.4%	12, 4.8%	0.018
Hypoglycemic, n (%)	35, 7.0%	23, 9.2%	12, 4.8%	0.055
Body mass index (kg/m^2^)	26.52 ± 4.23	26.90 ± 3.54	26.14 ± 4.79	0.044
Waist circumference (cm)	93.33 ± 12.01	98.76 ± 9.65	87.93 ± 11.70	<0.001
Obesity, n (%)	94, 18.8%	42, 16.9%	52, 20.6%	0.304
Abdominal obesity, n (%)	193, 38.6%	78, 31.3%	115, 45.8%	0.001
*Structure—vascular function and aging*				
Intima–media thickness (mm)	0.682 ± 0.109	0.699 ± 0.116	0.665 ± 0.100	0.001
cfPWV (m/s)	6.53 ± 2.03	6.86 ± 2.20	6.21 ± 1.79	<0.001
Vascular aging index	61.23 ± 12.86	63.47 ± 13.75	59.04 ± 11.54	<0.001

The values are displayed as means ± standard deviations for continuous data and as numbers and proportions for categorical data. MD, Mediterranean diet; g/W, grams/week; h/W, hours/week; LDL, low-density lipoprotein; HDL, high-density lipoprotein; HbA1c, glycosylated hemoglobin; cfPWV, femoral carotid pulse wave velocity. *p*: differences between men and women.

**Table 2 nutrients-16-02565-t002:** Lifestyles, risk factors, and vascular structure and function according to vascular aging.

Lifestyles	HVA (94, 18.9%)	Non HVA (407, 81.1%)	*p*
MD score	6.91 ± 2.20	7.20 ± 2.05	0.226
Adherence to MD, n (%)	33, 35.1%	180, 44.6%	0.106
Alcohol consumption (g/W)	34.89 ± 57.99	41.94 ± 64.46	0.331
Adequate alcohol consumption, n (%)	47, 50.0%	202, 50.0%	1.000
Years of smoking	12.45 ± 15.41	12.90 ± 17.70	0.820
Never smoked, n (%)	65, 69.1%	299, 67.0%	0.037
Total physical activity (h/W)	29.19 ± 9.79	26.58 ± 9.41	0.017
More than 26 h/W, n (%)	56, 60.2%	188, 47.1%	0.028
Sedentary time (h/W)	138.57 ± 9.81	141.28 ± 9.44	0.014
Less than 142 h/W, n (%)	57, 61.3%	194, 48.6%	0.029
Number of healthy lifestyles	2.74 ± 1.20	2.63 ± 1.32	0.444
More than 2 healthy lifestyles, n (%)	57, 60.6%	195, 48.3%	0.039
*Conventional risk factors*			
Age (years)	52.64 ± 13.29	56.66 ± 14.34	0.010
Systolic blood pressure (mmHg)	109.59 ± 12.61	123.32 ± 24.28	<0.001
Diastolic blood pressure (mmHg)	71.10 ± 8.28	76.56 ± 10.24	<0.001
Mean arterial pressure (mmHg)	83.93 ± 9.00	92.14 ± 12.87	<0.001
Pulse pressure (mmHg)	38.48 ± 8.90	46.76, 21.30	<0.001
Hypertensive, n (%)	0, 0.0%	147, 36.4%	<0.001
Antihypertensive, n (%)	0, 0.0%	96, 19.3%	<0.001
Total cholesterol (mg/dL)	193.27 ± 32.39	194.95 ± 32.50	0.651
LDL cholesterol (mg/dL)	113.42 ± 28.02	115.87 ± 29.72	0.469
HDL cholesterol (mg/dL)	62.42 ± 18.00	57.85 ± 15.62	0.014
Triglycerides (mg/dL)	83.33 ± 32.99	107.66 ± 55.97	<0.001
Atherogenic index	3.32 ± 1.09	3.59 ± 1.06	0.027
Dyslipidemic	29, 30.9%	160, 39.6%	0.126
Lipid-lowering, n (%)	13, 13.8%	88, 21.8%	0.086
Plasma glucose (mg/dL)	83.41 ± 10.27	89.41 ± 18.49	<0.001
HbA1c (%)	5.30 ± 0.29	5.53 ± 0.59	<0.001
Diabetes mellitus type 2, n (%)	0, 0.0%	38, 9.4%	0.002
Hypoglycemic, n (%)	0, 0.0%	35, 7.0%	0.003
Body mass index (kg/m^2^)	24.43 ± 3.26	27.00 ± 4.29	<0.001
Waist circumference (cm)	88.48 ± 9.41	94.45 ± 12.28	<0.001
Obesity, n (%)	6, 6.4%	88, 21.8%	0.001
Abdominal obesidad, n (%)	21, 22.3%	171, 42.4%	<0.001
*Structure—vascular function and aging*			
Intima–media thickness (mm)	0.62 ± 0.08	0.70 ± 0.11	<0.001
cfPWV (m/s)	4.83 ± 0.75	6.93 ± 2.02	<0.001
Vascular aging index	50.51 ± 5.67	63.74 ± 12.78	<0.001

The values are displayed as means ± standard deviations for continuous data and as numbers and proportions for categorical data. MD, Mediterranean diet; g/W, grams/week; h/W, hours/week; LDL, low-density lipoprotein; HDL, high-density lipoprotein; HbA1c, glycosylated hemoglobin; HVA, healthy vascular aging; cfPWV, femoral carotid pulse wave velocity. *p*: Differences between HVA and non HVA.

**Table 3 nutrients-16-02565-t003:** Correlation between index vascular aging, cfPWV and intima–media thickness with lifestyles.

VAI	Overall (501)	Men (249)	Women (251)
MD score	−0.102 *	−0.082 *	0.058
Alcohol consumption, g/W	0.228 **	0.221 **	−0.040
Year smoking	0.092 *	0.105	0.010
Total physical activity (h/W)	−0.158 *	−0.120	−0.161 *
Sedentary time (h/W)	0.165 **	0.135 *	−0.162 *
Number of healthy lifestyles	−0.199 **	−0.197 **	−0.063
cfPWV (m/s)			
MD score	−0.082	−0.048	0.075
Alcohol consumption, g/W	0.145 **	0.187 *	−0.031
Year smoking	0.082	0.099	0.026
Total physical activity (h/W)	−0.174 **	−0.155 *	−0.159 *
Sedentary time (h/W)	0.181 **	0.170 *	−0.160 *
Number of healthy lifestyles	−0.178 **	−0.173 *	−0.076
Intima–media thickness (mm)			
MD score	−0.082	−0.047	−0.058
Alcohol consumption, g/W	0.209 **	0.159 *	0.120 *
Year smoking	0.074	0.042	0.088
Total physical activity (h/W)	0.013	0.082	0.054
Sedentary time (h/W)	0.016	0.076	0.057
Number of healthy lifestyles	0.080	−0.012	−0.046

Age-adjusted Pearson correlation. MD, Mediterranean diet; g/W, grams/week; VAI, vascular aging index; cfPWV, femoral carotid pulse wave velocity. Pearson coefficient. * *p* < 0.05; ** *p* < 0.01.

**Table 4 nutrients-16-02565-t004:** Association between vascular aging and lifestyles. Multiple regression analysis.

VAI	β	(95% CI)	*p*
Overall			
MD score	−0.056	−0.418–0.307	0.763
Alcohol consumption, g/W	0.020	0.008–0.032	0.001
Year smoking	0.028	−0.013–0.069	0.185
Total physical activity (h/W)	−0.102	−0.176–−0.028	0.007
Sedentary time (h/W)	0.109	0.036–0.183	0.004
Number of healthy lifestyles	−0.640	−1.195–−0.086	0.024
Men			
MD score	−0.044	−0.598–0.509	0.875
Alcohol consumption, g/W	0.020	0.005–0.034	0.008
Year smoking	0.045	−0.014–0.103	0.133
Total physical activity (h/W)	−0.096	−0.203–0.011	0.080
Sedentary time (h/W)	0.108	0.001–0.214	0.048
Number of healthy lifestyles	−1.054	−1.741–−0.367	0.003
Women			
MD score	−0.010	−0.459–0.439	0.965
Alcohol consumption, g/W	−0.005	−0.027–0.018	0.690
Year smoking	−0.023	−0.080–0.033	0.416
Total physical activity (h/W)	−0.099	−0.195–−0.002	0.044
Sedentary time (h/W)	0.099	0.195–0.003	0.043
Number of healthy lifestyles	−0.256	−0.820–0.309	0.373

Multiple regression: dependent variables (estimated vascular aging with VAI). Lifestyle independent variables. Adjustment variables: age, mean arterial pressure, atherogenic index, body mass index and glycosylated hemoglobin. MD, Mediterranean diet; g/W, grams/week; VAI, vascular aging index.

**Table 5 nutrients-16-02565-t005:** Association between healthy vascular aging and lifestyles. Logistics regression analysis.

VAI	OR	(95% CI)	*p*
Overall			
MD adherence	0.571	0.333–0.981	0.042
AA consumption	0.993	0.606–1.627	0.977
Smoker	0.648	0.375–1.119	0.119
PAT > 26 h/W	1.735	1.048–2.871	0.032
ST—142 h/W	1.696	1.025–2.805	0.040
>2 HL	1.877	1.123–3.136	0.016
Men			
MD adherence	0.413	0.170–1.004	0.051
AA consumption	0.852	0.425–1.708	0.653
Smoker	0.382	0.162–0.902	0.028
PAT > 26 h/W	1.825	0.878–3.793	0.107
ST—142 h/W	1.738	0.838–3.604	0.136
>2 HL	1.998	0.956–4.175	0.066
Women			
MD adherence	0.766	0.377–1.558	0.462
AA consumption	0.968	0.465–2.014	0.931
Smoker	0.812	0.381–1.731	0.589
PAT > 26 h/W	1.668	0.817–3.404	0.160
ST—142 h/W	1.655	0.810–3.379	0.167
>2 HL	1.769	0.848–3.692	0.128

Multiple regression: dependent variables (estimated vascular aging with VAI). Lifestyle independent variables. Adjustment variables: age, mean arterial pressure, atherogenic index, body mass index and glycosylated hemoglobin. MD, Mediterranean diet; g/W, grams/week; AA consumption, adequate alcohol consumption; PAT H/W, physical activity total hours/week; ST H/W, sedentary hours per week; HL, number of healthy lifestyles; VAI, vascular aging index.

## Data Availability

Data supporting the findings of this study are available from the corresponding author upon reasonable request. The data are not publicly available due to privacy.

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
