# Peer review of "The Relationship between Healthy Vascular Aging with the Mediterranean Diet and Other Lifestyles in the Spanish Population: The EVA Study"

_nutrients, 2024, doi:10.3390/nu16152565_

Round 1
Reviewer 1 Report
Comments and Suggestions for Authors
This is an interesting study evaluating the role of mediterranean type of diet on atherosclerosis process.
The authors use a score for Mediterranean diet. As they include a population withna large variance of age, the diet score they used should be validedated for this age range. Furthermore, people in Spain have also other dietary habits like various types of alcohol orlocal foods. Those dietary habits cannot be represented through a diet score, so I would recommend to make at least a comment for this. Several other risk factors should be carefully addressed as they consist co-founders in this analysis; like Lpa, non-HDL. Furthermore it would be interesting to gve a correlation between SCORE (atherosclerosis) and carotid disease and how Med Diet can modify ( in tertiles of diet score)
Author Response
Review 1
( ) I would not like to sign my review report
(x) I would like to sign my review report
Quality of English Language
( ) am not qualified to assess the quality of English in this paper
( ) English very difficult to understand/incomprehensible
( ) Extensive editing of English language required
( ) Moderate editing of English language required
( ) Minor editing of English language required
(x) English language fine. No issues detected
|
|
||||
|
Yes |
Can be improved |
Must be improved |
Not applicable |
|
|
Does the introduction provide sufficient background and include all relevant references? |
( ) |
(x) |
( ) |
( ) |
|
Is the research design appropriate? |
(x) |
( ) |
( ) |
( ) |
|
Are the methods adequately described? |
(x) |
( ) |
( ) |
( ) |
|
Are the results clearly presented? |
(x) |
( ) |
( ) |
( ) |
|
Are the conclusions supported by the results? |
( ) |
(x) |
( ) |
( ) |
|
Yes |
Can be improved |
Must be improved |
Not applicable |
|
|
Does the introduction provide sufficient background and include all relevant references? |
( ) |
(x) |
( ) |
( ) |
|
Is the research design appropriate? |
(x) |
( ) |
( ) |
( ) |
|
Are the methods adequately described? |
(x) |
( ) |
( ) |
( ) |
|
Are the results clearly presented? |
(x) |
( ) |
( ) |
( ) |
Comments and Suggestions for Authors
This is an interesting study evaluating the role of mediterranean type of diet on atherosclerosis process.
Authors' Answer
Thank you for reviewing the manuscript and the positive comment on it.
1.- The authors use a score for Mediterranean diet. As they include a population with a large variance of age, the diet score they used should be validated for this age range.
Authors' Answer
The MEDAS questionnaire was valid in 2011 in 7447 asymptomatic participants at high risk of coronary heart disease aged between 55 and 80 years in 10 primary care centers in Spain [1]. It has been validated again in 2021 in 6760 people from the PREDIMED-Plus study aged between 55-75 years to analyse the validity of screening for adherence to the Mediterranean diet with calorie restriction [2]. It has also been validated in other towns in England [3] or Germany [4]
However, it is true that as far as we know it is not validated for the range of population included in this study.
Therefore, to give more clarity to the manuscript we have added the following in the current version of the manuscript:
2.3.1. Healthy lifestyles
2.3.1.1 Mediterranean diet
Adherence to the MD was assessed using the PREDIMED study questionnaire with 7447 asymptomatic participants aged between 55 and 80 years from the Spanish population with high risk of coronary heart disease [1]. It consists of 14 items, of which 12 ask about the frequency of food consumption and two about the typical eating habits of the Spanish population. It was validated again in 2021 with 6760 subjects in the PREDIMED-Plus study (n = 6760, 55-75 years) [2], but it has not been validated for the population aged under 55 years.
Limitations and strengths
This study has a number of limitations and strengths. The main limitation is that the analysis is based on cross-sectional data, which prevents us from establishing causality. Another limitation is that three of the four lifestyles analyzed were recorded subjectively through questionnaires, and the MD adherence questionnaire has not been validated in the Spanish population under 55 years of age. Among the strengths of the study are sample selection through random sampling stratified by age and sex groups from a reference population of more than 43,000 subjects, while the measurement of IMT of carotid arteries and cfPWV were performed under standardized conditions, by experienced evaluators and with validated devices. Furthermore, all analytical measurements were carried out in laboratories with adequate quality controls.
2.- Furthermore, people in Spain have also other dietary habits like various types of alcohol or local foods.
Authors' Answer
As suggested by the reviewer, the Mediterranean diet pattern includes the foods eaten in 14 items, within the alcohol intake only includes red wine and not another type of alcohol. In addition, it does not measure micronutrients or vitamins that can influence vascular aging. However, we cannot forget that the objective of this study is: The aim of this study was to analyze the relationship between healthy vascular aging (HVA) and the Mediterranean diet alongside other lifestyles in a Spanish population aged 35 to 75 years without previous cardiovascular diseases. These brief diet assessment tools can be useful for estimating food intake and diet quality in large-scale, time-constrained epidemiological studies.
3.- Those dietary habits cannot be represented through a diet score, so I would recommend to make at least a comment for this.
Authors' Answer
The MEDAS questionnaire, as we explained in methods, gives a score to each of the 14 questions asked and with them we can make a total score of it. However, there is no consensus to establish the cut-off point from which we can consider good or bad adherence. Therefore, in this work we have used the median value, as we have done with other lifestyles to classify subjects as adherent or non-adherent.
We have added the following sentence to the manuscript:
2.3.1. Healthy lifestyles
2.3.1.1 Mediterranean diet
Adherence to the MD was assessed using the PREDIMED study questionnaire [38]. Adherence to the MD was assessed using the PREDIMED study questionnaire with 7447 asymptomatic participants aged between 55 and 80 years from the Spanish population with high risk of coronary heart disease [1]. It consists of 14 items, of which 12 ask about the frequency of food consumption and two about the typical eating habits of the Spanish population. It was validated again in 2021 with 6760 subjects in the PREDIMED-Plus study (n = 6760, 55-75 years) [2], but it has not been validated for the population aged under 55 years. Each question is scored zero or one, with one point scored for: use of olive oil as the main cooking fat, daily consumption of four or more tablespoons of olive oil (one tablespoon = 13.5 g), two or more servings of vegetables, three or more pieces of fruit, less than one serving of red or processed meat, less than one serving of animal fat, less than a 100 ml cup of sugary drinks, eating white meat in greater proportion to red meat, weekly consumption of seven or more glasses of wine, three or more servings of legumes, three or more servings of fish, three or more servings of nuts or dried fruits, two or more servings of sofrito (home-made sauce of onions and/or garlic and tomato, slow-fried in extra-virgin olive oil) and less than two baked goods. The final score ranges between 0 and 14 points, with MD adherence considered for a score above the median (7 or more points) [38]. There is no consensus on this cut-off point, which thus represents a limitation.
4.- Several other risk factors should be carefully addressed as they consist co-founders in this analysis; like Lpa, non-HDL.
Authors' Answer
In this work, we have chosen age and the different cardiovascular risk factors as adjustment variables, considering the one that can best represent each of them, such as: For hypertension: mean arterial pressure; for dyslipidemia: atherogenic index; for type 2 diabetes mellitus: HbA1c and for obesity: body mass index, we have not used any more adjustment variables to avoid interaction effects between them. Choosing those with values of p <0.05 in table 2 of the manuscript.
On the other hand, in this work we have not included the value of the LPA.
For a larger explanation of the manuscript we have added the following paragraph at the bottom of Figure 4:
Figure 4. Differences in VAI scores between subjects with and without healthy lifestyles, overall and by sex. Healthy lifestyles considered to be: a) adherence to the Mediterranean diet score greater than 7. b) no alcohol consumption; c) Adequate alcohol consumption for women was considered to be <140 g/week and for men <210 g/week; d) more than 26 hours a week of physical activity; e) less than 142 hours a week sitting and f) more than two healthy lifestyles.
MD, Mediterranean diet; AA, adequate alcohol consumption; PAT h/W, physical activity total hours/week; S h/W, sedentary hours per week; HL, number of healthy lifestyles; VAI, vascular aging index. Brown represents the overall results, blue the results for men and pink the results for women. * reflects whether the difference between subjects with and without healthy lifestyles is significant.
Similarly, in the section on statistical analysis, we have added:
We considered a healthy lifestyle to be when subjects had: a) an adherence to the Mediterranean diet score greater than 7; b) Adequate alcohol consumption for women was considered to be <140 g/week and for men <210 g/week;; c) no smoking; d) more than 26 hours a week of physical activity; e) less than 142 hours a week of sedentary time and f) more than two healthy lifestyles.
5.- Furthermore it would be interesting to have a correlation between SCORE (atherosclerosis) and carotid disease and how
Además, sería interesante establecer una correlación entre el SCORE (aterosclerosis) y la enfermedad carotídea
Authors' Answer
Following the reviewer's indications, we have included in Table 3 the correlation of lifestyles with VAI, cfPWV and intimate mean thickness. Table 3 remains in the current version as follows:
Table 3. Correlation between index vascular aging, cfPWV and intima media thickness with lifestyles
|
VAI |
Global (501) |
Men (249) |
Women (251) |
|
Score the MD |
-0.102* |
-0.082* |
0.058 |
|
Alcohol consumption, g/W |
0.228** |
0.221** |
-0.040 |
|
Year smoking |
0.092* |
0.105 |
0.010 |
|
Total physical activity, (h/W) |
-0.158* |
-0.120 |
-0.161* |
|
Sitting Time, (h/W) |
0.165** |
0.135* |
-0.162* |
|
Number of healthy lifestyles |
-0.199** |
-0.197** |
-0.063 |
|
cfPWV, (m/sec) |
|
|
|
|
Score the MD |
-0.082 |
-0.048 |
0.075 |
|
Alcohol consumption, g/W |
0.145** |
0.187* |
-0.031 |
|
Year smoking |
0.082 |
0.099 |
0.026 |
|
Total physical activity, (h/W) |
-0.174** |
-0.155* |
-0.159* |
|
Sitting Time, (h/W) |
0.181** |
0.170* |
-0.160* |
|
Number of healthy lifestyles |
-0.178** |
-0.173* |
-0.076 |
|
Intima Media Thickness, (mm) |
|
|
|
|
Score the MD |
-0.082 |
-0.047 |
-0.058 |
|
Alcohol consumption, g/W |
0.209** |
0.159* |
0.120* |
|
Year smoking |
0.074 |
0.042 |
0.088 |
|
Total physical activity, (h/W) |
0.013 |
0.082 |
0.054 |
|
Sitting Time, (h/W) |
0.016 |
0.076 |
0.057 |
|
Number of healthy lifestyles |
0.080 |
-0.012 |
-0.046 |
Correlation de Pearson Age-adjusted.
MD, Mediterranean diet; g/W, grams/week; VAI, Vascular Aging Index; cfPWV, femoral carotid pulse wave velocity.
Pearson Coefficient. * p < 0.05; ** p < 0.01.
We have modified in the text
3.3. Correlation of vascular aging with Mediterranean diet and other lifestyles
Table 3 shows the correlation of the VAI, cfPWV and cIMT with lifestyles adjusted by age, overall and by sex. The VAI showed a negative association with the mean value of MD (r=-0.102), hours of total physical activity (r=-0.158) and the number of healthy lifestyles (r=-0.199), and a positive association with consumption of alcohol (r=0.228), years of smoking (r=0.092) and sedentary time (r=0.165).
6.- Med Diet can modify (in tertiles of diet score)
Authors' Answer
Following the reviewer's indications, we have segmented adherence to the Mediterranean diet by tertiles. The results are shown in the table below.
Lifestyles, Risk Factors, and Vascular Structure and Function according to vascular aging.
|
|
HVA |
Non HVA 6.94 |
p |
||
|
MD Tertil 1 |
4.00 |
±0.87 |
4.45 |
±0.77 |
0.017 |
|
MD Tertil 2 |
6.94 |
±0.75 |
7.03 |
±0.82 |
0.245 |
|
MD Tertil 3 |
9.80 |
±0.98 |
10.05 |
±1.05 |
0.156 |
As we can see, the results are similar and the subjects classified as non-HVA have higher DM score values, although it only reaches significance in tertile 1.
Since we have classified the rest of the lifestyles into two categories, we think that we should follow the same criteria in the DM, for a better understanding of the manuscript.
References
- Schröder, H.; Fitó, M.; Estruch, R.; Martínez-González, M.A.; Corella, D.; Salas-Salvadó, J.; Lamuela-Raventós, R.; Ros, E.; Salaverría, I.; Fiol, M., et al. A short screener is valid for assessing Mediterranean diet adherence among older Spanish men and women. J Nutr 2011, 141, 1140-1145, doi:10.3945/jn.110.135566.
- Schröder, H.; Zomeño, M.D.; Martínez-González, M.A.; Salas-Salvadó, J.; Corella, D.; Vioque, J.; Romaguera, D.; Martínez, J.A.; Tinahones, F.J.; Miranda, J.L., et al. Validity of the energy-restricted Mediterranean Diet Adherence Screener. Clin Nutr 2021, 40, 4971-4979, doi:10.1016/j.clnu.2021.06.030.
- Papadaki, A.; Johnson, L.; Toumpakari, Z.; England, C.; Rai, M.; Toms, S.; Penfold, C.; Zazpe, I.; Martínez-González, M.A.; Feder, G. Validation of the English Version of the 14-Item Mediterranean Diet Adherence Screener of the PREDIMED Study, in People at High Cardiovascular Risk in the UK. Nutrients 2018, 10, doi:10.3390/nu10020138.
- Hebestreit, K.; Yahiaoui-Doktor, M.; Engel, C.; Vetter, W.; Siniatchkin, M.; Erickson, N.; Halle, M.; Kiechle, M.; Bischoff, S.C. Validation of the German version of the Mediterranean Diet Adherence Screener (MEDAS) questionnaire. BMC Cancer 2017, 17, 341, doi:10.1186/s12885-017-3337-y.

Reviewer 2 Report
Comments and Suggestions for Authors
The authors conducted an observational study examining the association between Mediterranean diet adherence and other lifestyles with healthy vascular aging in adults. By analyzing the data of 501 participants of the EVA study, the authors showed that alcohol consumption, physical activity, and sitting time, but not Mediterranean diet adherence, were each associated with the vascular aging index. Physical activity of greater than 26 hours per week and sitting time of less than 142 hours per week were each associated with greater odds of having healthy vascular aging. Interestingly, Mediterranean diet adherence was associated with a lower odds of healthy vascular aging. There are some comments.
1. Introduction: "- the MESA study demonstrated that continuous smoking over a period of 10 years was linked to a greater increase in the progression of cfPWV, compared to nonsmokers." (Lines 122-123 on page 3) However, according to the cited study (reference 31), smoking was associated with an increase in the aortic arch pulse-wave velocity, not the carotid-femoral pulse-wave velocity. A more precise description is recommended.
2. Introduction: "This increase only occurred in men but not in women." (Line 124 on page 3) However, the cited study (reference 31) did not show that the increase occurred only in men but not women.
3. Materials and Methods (Statistical Analysis): Multiple linear regression analysis was applied. (Line 262 on page 6) Please describe whether (and how) the assumptions, such as constant variance and the absence of outliers' effects, were checked and met.
4. Materials and Methods (Statistical Analysis): "- age, sex, mean arterial pressure, atherogenic index, HbA1c and body mass index were included as adjustment variables." (Line 272 on page 6) However, it is unclear why the authors chose these variables as potential confounders. In fact, as shown in Table 2, adults with healthy vascular aging showed significantly different levels of systolic and diastolic blood pressure, plasma glucose, and waist circumference, in addition to mean blood pressure, HbA1c, and body mass index, as compared with those without healthy vascular aging.
5. Results (Figures 2 and 3): Please consider presenting the results using a Pie chart if the primary goal of the figures was to show the proportions.
6. Results (Figure 4): It seems that the reference groups were "without MD adherence," "without AA consumption," "nonsmoker," "PAT <= 26h/W", "S >= 142 h/W", "<= 2HL". Please also show these reference groups in this figure. Also, based on the figure and the description in the footnote, it is still unclear what the comparison of the "*" mark actually referred to.
7. Results (Line 367 on page 12): "compared to non-HVA, subjects classified as HVA were more likely to have Mediterranean diet adherence (OR = 0.571; 95% CI: 0.333 to 0.981), -." However, according to the logistic regression model and the estimated OR, which was 0.571 (0.333 to 0.981), the description should be "compared to Mediterranean diet non-adherence, Mediterranean diet adherence was associated with a lower odd (likelihood) of being classified as HVA, relative to non-HVA (OR = 0.571; 95% CI: 0.333 to 0.981), -." A revision is recommended. Likewise, the rest of the description of the logistic regression results (lines 366-373) should be revised. Corresponding descriptions in the ABSTRACT should also be revised.
8. Discussion: In contrast to previous studies, the authors showed that Mediterranean diet adherence was associated with lower odds of healthy vascular aging. A discussion of this paradoxical finding is recommended.
9. Discussion: As mentioned in the Introduction (Line 140 on page 3), jointly analyzing the effects of different lifestyles is a uniqueness of this study. I recommend a discussion of the findings regarding the number of healthy lifestyles. Of note, the authors observed an association of the number of healthy lifestyles with vascular aging index and healthy vascular aging, as shown in Tables 4 and 5. However, the association was less significant in women. A discussion of this finding is recommended.
10. Discussion: In this study, confounding is a major source of biased observations. Although the analysis was adjusted for some covariates, the results were possibly biased due to residual confounding by unmeasured covariates. A discussion is recommended.
Comments on the Quality of English Language
Moderate editing of the English language is required.
Author Response
Review 2
(x) I would not like to sign my review report
( ) I would like to sign my review report
Quality of English Language
( ) I am not qualified to assess the quality of English in this paper
( ) English very difficult to understand/incomprehensible
( ) Extensive editing of English language required
(x) Moderate editing of English language required
( ) Minor editing of English language required
( ) English language fine. No issues detected
|
Authors' Answer El manuscrito ha sido editado por un native ingles. Adjuntamos certificado de edición del mismo |
|||||||||
|
Yes |
Can be improved |
Must be improved |
Not applicable |
|
|||||
|
Does the introduction provide sufficient background and include all relevant references? |
( ) |
(x) |
( ) |
( ) |
|
||||
|
Is the research design appropriate? |
(x) |
( ) |
( ) |
( ) |
|
||||
|
Are the methods adequately described? |
( ) |
(x) |
( ) |
( ) |
|
||||
|
Are the results clearly presented? |
( ) |
(x) |
( ) |
( ) |
|
||||
|
Are the conclusions supported by the results? |
(x) |
( ) |
( ) |
( ) |
|
||||
Comments and Suggestions for Authors
The authors conducted an observational study examining the association between Mediterranean diet adherence and other lifestyles with healthy vascular aging in adults. By analyzing the data of 501 participants of the EVA study, the authors showed that alcohol consumption, physical activity, and sitting time, but not Mediterranean diet adherence, were each associated with the vascular aging index. Physical activity of greater than 26 hours per week and sitting time of less than 142 hours per week were each associated with greater odds of having healthy vascular aging. Interestingly, Mediterranean diet adherence was associated with a lower odds of healthy vascular aging. There are some comments.
Authors' Answer
Thank you for reviewing the manuscript and for your comments, which will certainly improve the quality of the manuscript.
- Introduction: "- the MESA study demonstrated that continuous smoking over a period of 10 years was linked to a greater increase in the progression of cfPWV, compared to nonsmokers." (Lines 122-123 on page 3) However, according to the cited study (reference 31), smoking was associated with an increase in the aortic arch pulse-wave velocity, not the carotid-femoral pulse-wave velocity. A more precise description is recommended.
Authors' Answer
We have fixed the bug remaining in the new version as follows:
Thus, the MESA study demonstrated that continuous smoking over a period of 10 years was associated with an increase in the aortic arch pulse-wave velocity, compared to non-smokers.
- Introduction: "This increase only occurred in men but not in women." (Line 124 on page 3) However, the cited study (reference 31) did not show that the increase occurred only in men but not women.
Authors' Answer
We have fixed the bug remaining in the new version as follows:
This increase occurred in men and in women.
- Materials and Methods (Statistical Analysis): Multiple linear regression analysis was applied. (Line 262 on page 6) Please describe whether (and how) the assumptions, such as constant variance and the absence of outliers' effects, were checked and met.
Authors' Answer
We have added the following information in the statistical analysis.
2.4. Statistical Analysis
Data are presented using means ± standard deviations and numbers or percentages depending on whether variables are continuous or categorical. The comparison between men and women was done with chi-square tests for percentages and Student's t tests for continuous variables. The Pearson correlation coefficient was used to analyze the relationship between continuous variables.
To analyze the association between the average VAI score and the different lifestyles, six multiple linear regression models were used, using the VAI as dependent variable and the number of healthy lifestyles, average MD adherence score, weekly alcohol consumption in g/week, years of smoking, number of active hours per week and number of sedentary hours per week as independent variables.
The constant variance hypothesis was verified with variance homogeneity tests using the Levene statistic in all variables analyzed. Using the classification as HVA or non- HVA as the dependent variable, the different lifestyles as independent variables and the age and cardiovascular risk factors (mean arterial pressure, atherogenic index, HbA1c and body mass index) as adjustment variables, the p value was >0.050 in all the variables. The outlier values were analyzed with box-whisker plots, the number of values was not excluded from the analysis since in those variables with outlier values, the number is small and within biological values.
To analyze the association between HVA individuals and different lifestyles, six logistic regression models were used. Those classified as HVA or not HVA were dependent variables (coded HVA=1, No HVA=0), healthy lifestyles (coded Yes=1, No=0) were independent variables. The median score for the different healthy lifestyles, except in the case of alcohol, which was used if they had adequate alcohol consumption or not (considered to be <140 g/week in women and 210 g/week in men).
In all models, age, sex, mean arterial pressure, atherogenic index, HbA1c and body mass index were included as adjustment variables.
All analyses were performed globally and by sex. The SPSS Statistics program for Windows, version 28.0 (IBM Corp, Armonk, NY, USA) was used. A value of p < 0.05 was considered the statistical significance limit.
- Materials and Methods (Statistical Analysis): "- age, sex, mean arterial pressure, atherogenic index, HbA1c and body mass index were included as adjustment variables." (Line 272 on page 6) However, it is unclear why the authors chose these variables as potential confounders. In fact, as shown in Table 2, adults with healthy vascular aging showed significantly different levels of systolic and diastolic blood pressure, plasma glucose, and waist circumference, in addition to mean blood pressure, HbA1c, and body mass index, as compared with those without healthy vascular aging.
Authors' Answer
In this work, we have chosen age, sex and the different cardiovascular risk factors as adjustment variables, considering the parameter that can best represent each of them, such as: For hypertension: mean arterial pressure; for dyslipidemia: atherogenic index; for type 2 diabetes mellitus: HbA1c and for obesity: body mass index, we have not included more adjustment variables to avoid interaction effects between them and possible overadjustment. Choosing those with values of p <0.05 in table 2 of the manuscript.
We have added the following sentence in the statistical analysis section:
In all models, age, sex, mean arterial pressure, atherogenic index, HbA1c and body mass index were included as adjustment variables. The selection criteria for these were: one for each cardiovascular risk factor with a p value <0.05 in Table 2 and which combined the largest number of variables related to said factor.
- Results (Figures 2 and 3): Please consider presenting the results using a Pie chart if the primary goal of the figures was to show the proportions.
Authors' Answer
Following their recommendations, we have presented the results of figures 2 and 3 using a pie chart.
It remains in the current version as follows:
Figure 2.
Figure 3.
- Results (Figure 4): It seems that the reference groups were "without MD adherence," "without AA consumption," "nonsmoker," "PAT <= 26h/W", "S >= 142 h/W", "<= 2HL". Please also show these reference groups in this figure. Also, based on the figure and the description in the footnote, it is still unclear what the comparison of the "*" mark actually referred to.
Authors' Answer
Figure 4 shows the differences in the mean value of the VAI between subjects who have a healthy lifestyle and those who do not have a healthy lifestyle.
We consider healthy lifestyles 1. Adherence to the Mediterranean Mediterranean diet. 2. Adequate alcohol consumption for women was considered to be <140 g/week and for men <210 g/week;. 3.- Do not smoke. 4.- More than 26 hours a week of physical activity. 5.- Less than 142 hours a week sitting and 6.- Have less than two healthy lifestyles.
The * reflects whether the difference between subjects with and without healthy lifestyles is significant
Following the recommendations of the reviewer we have added in the caption:
Figure 4. Differences in VAI scores between subjects with and without healthy lifestyles, overall and by sex. Healthy lifestyles considered to be: a) adherence to the Mediterranean diet score greater than 7. b) Adequate alcohol consumption for women was considered to be <140 g/week and for men <210 g/week; c) no smoking; d) more than 26 hours a week of physical activity; e) less than 142 hours a week sitting and f) more than two healthy lifestyles.
MD, Mediterranean diet; AA, adequate alcohol consumption; PAT h/W, physical activity total hours/week; S h/W, sedentary hours per week; HL, number of healthy lifestyles; VAI, vascular aging index. Brown represents the overall results, blue the results for men and pink the results for women. * reflects whether the difference between subjects with and without healthy lifestyles is significant.
Similarly, in the section on statistical analysis, we have added:
We considered a healthy lifestyle to be when subjects had: a) an adherence to the Mediterranean diet score greater than 7; b) Adequate alcohol consumption for women was considered to be <140 g/week and for men <210 g/week; c) no smoking; d) more than 26 hours a week of physical activity; e) less than 142 hours a week of sedentary time and f) more than two healthy lifestyles.
- Results (Line 367 on page 12): "compared to non-HVA, subjects classified as HVA were more likely to have Mediterranean diet adherence (OR = 0.571; 95% CI: 0.333 to 0.981), -." However, according to the logistic regression model and the estimated OR, which was 0.571 (0.333 to 0.981), the description should be "compared to Mediterranean diet non-adherence, Mediterranean diet adherence was associated with a lower odd (likelihood) of being classified as HVA, relative to non-HVA (OR = 0.571; 95% CI: 0.333 to 0.981), -." A revision is recommended. Likewise, the rest of the description of the logistic regression results (lines 366-373) should be revised. Corresponding descriptions in the ABSTRACT should also be revised.
Authors' Answer
We have rewritten the paragraph to the current version as follows:
3.5. Association between vascular aging index and healthy lifestyles. Logistic regression analysis
After adjustment for possible confounding factors, the logistic regression analysis yielded the following results: compared with non-adherence to the Mediterranean diet, adherence to the Mediterranean diet was associated with a higher likelihood of being classified as HVA, relative to non-HVA (OR = 0.571; 95% CI: 0.333 to 0.981); compared with less than 26 hours per week of physical activity, doing more than 26 hours per week of physical activity was associated with a higher likelihood of being classified as HVA, relative to non-HVA (OR = 1.735; 95% CI: 1.048 to 2.871); compared with more than 142 hours per week of sedentary time, spending less than 142 hours per week being sedentary was associated with a higher likelihood of being classified as HVA, relative to non-HVA (OR = 1.696; 95% CI: 1.025 to 2.805); and compared to having less than two healthy lifestyles, more than two healthy lifestyles was associated with a higher likelihood of being classified as HVA, compared to non-HVA (OR = 1.877; 95% CI: 1.123 to 3.136). In the analysis by sex, the results were similar, although there were no significant differences, except for smoking in men, as reflected in Table 5.
- Discussion: In contrast to previous studies, the authors showed that Mediterranean diet adherence was associated with lower odds of healthy vascular aging. A discussion of this paradoxical finding is recommended.
Authors' Answer
We have expanded this section of the discussion to the current manuscript as follows:
In summary, this paradoxical finding in our study may have several explanations: 1. The influence of the Mediterranean dietary pattern on HVA may not be so strong, with micronutrients and vitamins playing an important role, as shown by different studies carried out in animals or in humans [1-5]; 2. Most of the studies assessing the benefits of the Mediterranean diet have been carried out on specific diseases or factors and not on their importance in vascular aging; 3. The age range of our study’s population is broad and we should not forget that the MEDAS questionnaire has not been validated in people under 55 years of age, and there is no consensus on the cut-off points for good adherence; 4. Adherence to the Mediterranean diet is greater in older subjects, and the importance of age in aging is probably greater than that of the Mediterranean diet. Finally, there may be confounding factors that we have not considered in this work.
- Discussion: As mentioned in the Introduction (Line 140 on page 3), jointly analyzing the effects of different lifestyles is a uniqueness of this study. I recommend a discussion of the findings regarding the number of healthy lifestyles. Of note, the authors observed an association of the number of healthy lifestyles with vascular aging index and healthy vascular aging, as shown in Tables 4 and 5. However, the association was less significant in women. A discussion of this finding is recommended.
Authors' Answer
Siguiendo las indicaciones del revisor hemos incluido un párrafo en la discusión quedando en el manuscrito actual de la siguiente forma:
The results of this study suggest that the greater the number of healthy lifestyles, the greater the probability of being classified in the group of subjects with HVA and the lower the VAI value. This is likely due to the fact that such lifestyles mutually enhance the beneficial effect on HVA. Thus, the WHO considers that in order to achieve healthy aging, healthy lifestyle habits that reduce the development of pathologies associated with aging must be promoted in order to increase the quality of life of the aging population. It also recognizes that evidence is scarce, and that research on healthy aging should be promoted, with health systems adapting to the needs of older people, generating the human resources necessary to be able to carry out comprehensive care for older people [6]. In the review by Claas et al. [7], lifestyles such as adequate dietary intake and regular physical activity are seen as essential in primary prevention to avoid cardiovascular risk factors that will determine vascular aging. These measures, together with the absence of smoking and certain behavioral factors such as stress control and sleep duration, should be considered in lifestyle modification programs, and these healthy lifestyles should be implanted from childhood [8].
The differences found by sex have already been commented on in the discussion in the paragraph:
Finally, hormonal and non-hormonal factors influence differences between the sexes. The protection of endogenous estrogen until menopause in women is well known. Furthermore, in males, arterial stiffness increases linearly from puberty, which indicates that females intrinsically have stiffer main arteries than males, effects that are mitigated by sex steroids during reproductive life. Other factors, such as height, body fat distribution, and inflammatory factors may also play a role [9,10]
10.- Discussion: In this study, confounding is a major source of biased observations. Although the analysis was adjusted for some covariates, the results were possibly biased due to residual confounding by unmeasured covariates. A discussion is recommended.
Authors' Answer
Following the recommendations of the evaluator we have included the following paragraph in the discussion of the manuscript:
Nevertheless, we cannot forget that vascular aging is a complex phenomenon, in which many factors intervene, some of which are beyond our control, such as age, sex and the genetic load of each individual. Modifiable risk factors are those that we acquire throughout life, conditioned by different lifestyles that influence individual cardiovascular risk factors, psychological and inflammatory factors which cause an increase in oxidative stress and inflammation, leading to endothelial dysfunction. Thus, it is known that arterial aging is associated with changes that impact vascular function [11,12]. However, the extent to which these changes are produced by the action of environmental factors, lifestyle, psychological factors, inflammation or oxidative stress, predisposing each individual to mark a rhythm in his own vascular aging, has not been sufficiently researched. In this study we have tried to control for most of the confounding factors. Firstly, the sample was selected randomly, but it only represents the urban population. As adjustment variables for the cardiovascular risk factors, we tried to use those covariates that may best represent each of the cardiovascular risk factors, although we did not include inflammatory and psychological factors as possible confounding factors. Furthermore, three of the four lifestyles analyzed were recorded through questionnaires, so there may be some information bias. Finally, we must emphasize that physical activity and sedentary lifestyle, the only lifestyle objectively assessed with accelerometry, is the lifestyle that shows a clear and consistent association with vascular aging. For all these reasons, the results must be interpreted with caution, taking into account that there may be spurious associations that we have not considered.
References
- Remie, C.M.E.; Roumans, K.H.M.; Moonen, M.P.B.; Connell, N.J.; Havekes, B.; Mevenkamp, J.; Lindeboom, L.; de Wit, V.H.W.; van de Weijer, T.; Aarts, S., et al. Nicotinamide riboside supplementation alters body composition and skeletal muscle acetylcarnitine concentrations in healthy obese humans. Am J Clin Nutr 2020, 112, 413-426, doi:10.1093/ajcn/nqaa072.
- Brunt, V.E.; Gioscia-Ryan, R.A.; Casso, A.G.; VanDongen, N.S.; Ziemba, B.P.; Sapinsley, Z.J.; Richey, J.J.; Zigler, M.C.; Neilson, A.P.; Davy, K.P., et al. Trimethylamine-N-Oxide Promotes Age-Related Vascular Oxidative Stress and Endothelial Dysfunction in Mice and Healthy Humans. Hypertension 2020, 76, 101-112, doi:10.1161/hypertensionaha.120.14759.
- Ashor, A.W.; Shannon, O.M.; Werner, A.D.; Scialo, F.; Gilliard, C.N.; Cassel, K.S.; Seal, C.J.; Zheng, D.; Mathers, J.C.; Siervo, M. Effects of inorganic nitrate and vitamin C co-supplementation on blood pressure and vascular function in younger and older healthy adults: A randomised double-blind crossover trial. Clin Nutr 2020, 39, 708-717, doi:10.1016/j.clnu.2019.03.006.
- Martens, C.R.; Denman, B.A.; Mazzo, M.R.; Armstrong, M.L.; Reisdorph, N.; McQueen, M.B.; Chonchol, M.; Seals, D.R. Chronic nicotinamide riboside supplementation is well-tolerated and elevates NAD(+) in healthy middle-aged and older adults. Nat Commun 2018, 9, 1286, doi:10.1038/s41467-018-03421-7.
- Fontana, L. Interventions to promote cardiometabolic health and slow cardiovascular ageing. Nat Rev Cardiol 2018, 15, 566-577, doi:10.1038/s41569-018-0026-8.
- Rudnicka, E.; Napierała, P.; Podfigurna, A.; Męczekalski, B.; Smolarczyk, R.; Grymowicz, M. The World Health Organization (WHO) approach to healthy ageing. Maturitas 2020, 139, 6-11, doi:10.1016/j.maturitas.2020.05.018.
- Claas, S.A.; Arnett, D.K. The Role of Healthy Lifestyle in the Primordial Prevention of Cardiovascular Disease. Curr Cardiol Rep 2016, 18, 56, doi:10.1007/s11886-016-0728-7.
- Abrignani, M.G.; Lucà, F.; Favilli, S.; Benvenuto, M.; Rao, C.M.; Di Fusco, S.A.; Gabrielli, D.; Gulizia, M.M. Lifestyles and Cardiovascular Prevention in Childhood and Adolescence. Pediatr Cardiol 2019, 40, 1113-1125, doi:10.1007/s00246-019-02152-w.
- Kane, A.E.; Howlett, S.E. Differences in Cardiovascular Aging in Men and Women. Adv Exp Med Biol 2018, 1065, 389-411, doi:10.1007/978-3-319-77932-4_25.
- Keller, K.M.; Howlett, S.E. Sex Differences in the Biology and Pathology of the Aging Heart. Can J Cardiol 2016, 32, 1065-1073, doi:10.1016/j.cjca.2016.03.017.
- Laurent, S.; Boutouyrie, P.; Cunha, P.G.; Lacolley, P.; Nilsson, P.M. Concept of Extremes in Vascular Aging. Hypertension 2019, 74, 218-228, doi:10.1161/hypertensionaha.119.12655.
- Nowak, K.L.; Rossman, M.J.; Chonchol, M.; Seals, D.R. Strategies for Achieving Healthy Vascular Aging. Hypertension 2018, 71, 389-402, doi:10.1161/hypertensionaha.117.10439.
